# OPEN: Orthogonal Propagation with Ego-Network Modeling

**Liang Yang** [1,*], **Lina Kang** [1,*], **Qiuliang Zhang** [1,*], **Mengzhe Li** [1], **Bingxin Niu** [1],
**Dongxiao He** [2,†], **Zhen Wang** [3,4], **Chuan Wang** [5], **Xiaochun Cao** [6], **Yuanfang Guo** [7,8]

[1]School of Artificial Intelligence, Hebei University of Technology, Tianjin, China
[2]College of Intelligence and Computing, Tianjin University, Tianjin, China
[3] School of Artificial Intelligence, OPtics and ElectroNics (iOPEN),
Northwestern Polytechnical University, Xi'an, China
[4]School of Cybersecurity, Northwestern Polytechnical University, Xi'an, China
[5]State Key Laboratory of Information Security, IIE, CAS, Beijing, China
[6]School of Cyber Science and Technology, Sun Yat-sen University, Shenzhen, China
[7]School of Computer Science and Engineering, Beihang University, Beijing, China
[8]Zhongguancun Laboratory, Beijing, China
`yangliang@vip.qq.com, itkanglina@163.com, 3463194784@qq.com`
`limengzhefree@foxmail.com, niubingxin666@163.com, hedongxiao@tju.du.cn`
`w-zhen@nwpu.edu.cn,{wangchuan,caoxiaochun}@iie.ac.cn, andyguo@buaa.edu.cn`

## Abstract

To alleviate the unfavorable effect of noisy topology in Graph Neural networks (GNNs), some efforts perform the local topology refinement through the pairwise propagation weight learning and the multi-channel extension. Unfortunately, most of them suffer a common and fatal drawback: irrelevant propagation to one node and in multi-channels. These two kinds of irrelevances make propagation weights in multi-channels free to be determined by the labeled data, and thus the GNNs are exposed to overfitting. To tackle this issue, a novel Orthogonal Propagation with Ego-Network modeling (OPEN) is proposed by modeling relevances between propagations. Specifically, the relevance between propagations to one node is modeled by whole ego-network modeling, while the relevance between propagations in multi-channels is modeled via diversity requirement. By interpreting the propagations to one node from the perspective of dimension reduction, propagation weights are inferred from principal components of the ego-network, which are orthogonal to each other. Theoretical analysis and experimental evaluations reveal four attractive characteristics of OPEN as modeling high-order relationships beyond pairwise one, preventing overfitting, robustness, and high efficiency.

## 1 Introduction

Graph Neural Networks (GNNs) have been proven to be a powerful tool to explore irregular graph data by seamlessly combining graph topology and node attributes for node representation learning [1, 2]. From the perspective of spectral graph theory, GNNs, such as GCN [3] and ChebyNet [4], are proposed from graph signal filtering, whose filters are derived from the topology of whole graph, and their success is attributed to low-passing filtering. From the perspective of spatial propagation, GNNs, such as GraphSage [5] and MPNN [6], are presented by following the aggregation and combination scheme for node attributes smoothing [7]. Recent progress demonstrates the equivalence between

---

[*]Equal contribution.

[†]Corresponding author.

36th Conference on Neural Information Processing Systems (NeurIPS 2022).

these two perspectives [8]. However, both the filters based on graph topology and node attribute propagation between neighbourhoods take the implicit assumption that the graph topology is perfect and creditable [9]. Consequently, the oversmoothing issue [10, 11] and the expressive power loss [12], which are caused by overusing topology via stacking multiple layers, are identified.

Actually, the graph topology exists large amount of noises. Thus, graph topology refinement is critical to the accuracy and robustness of GNNs [13, 14, 9, 15]. The local refinement, especially through pairwise propagation weight learning, is widely investigated and employed due to the high accuracy and low computation load. As a representative local refinement, Graph Attention Network (GAT) [16] formulates the propagation weight on each edge as the attention between the two connected nodes, and introduces multi-channel propagation via multi-head attention to stabilize the learning process. Afterward, a number of approaches are proposed by following the scheme of propagation weight learning and multi-channel extension. For example, GaAN [17], PGCN [18], Masked-GCN [19] and DMP [20] extend GAT via different pairwise propagation weight learning functions. FAGCN [21] relaxes the non-negative constraint in GAT to a real number. ADSF [22] extends GAT to incorporate topology structure into the attention mechanism. GATv2 [23] removes the ranking irrelevance limit in GAT.

Unfortunately, most GNNs, especially those with local refinement, suffer a common but fatal drawback: **irrelevant propagations**. Firstly, the propagations to each node are irrelevant, since propagation weights are either predefined according to the topology or learned based on the contents of the two connected nodes. Actually, the propagations to one node should reflect global characteristics of its ego-network. Secondly, propagations in multi-channels are irrelevant, since parameters to channels are free to be learned without any specific constraints. Actually, to obtain stable and complementary representations from different channels, propagation schemes in different channels should be diverse [20]. These two kinds of irrelevances make propagation weights free to be determined by the labelled data for the specific task, thus resulting GNNs being susceptible to overfitting [24].

To tackle this issue, a novel Orthogonal Propagation with Ego-Network modeling (OPEN) is proposed by modeling these two kinds of relationships between propagations. Firstly, the relevance between propagations to one node is modeled by whole ego-network modeling. Specifically, by interpreting the propagations to one node from the perspective of dimension reduction, propagation weights are inferred from the principal component of ego-network, which corresponds to the mapping function maximizing the variance and captures the characteristics of the whole ego-network. Principal component can be obtained via the eigenvalue decomposition of the covariance matrix, which is efficient for the ego-network. Secondly, the propagations in multi-channels are implemented by employing *all* principal components, each of which corresponds to one channel. Since principal components are orthogonal to each other, the propagations in multi-channel are diverse. The proposed OPEN possesses some attractive characteristics: modeling high-order relationships beyond pairwise one, robustness, and high efficiency. Finally, theoretical analysis demonstrates that the two components of OPEN, i.e., Ego-Network modeling and Orthogonal Propagation in multi-channel, can prevent the over-smoothing issue.

The main contributions of this paper are summarized as follows:

- We investigate the common but fatal issue in GNNs, i.e., irrelevant propagation.
- We propose a novel Orthogonal Propagation with Ego-Network modeling (OPEN) by interpreting propagations in the ego-network from the perspective of dimension reduction.
- We provide a theoretical analysis of OPEN's capability on preventing oversmoothing issue.
- We experimentally verify the effectiveness and robustness of the proposed OPEN.

## 2   Preliminaries and Analysis

**Notations:** Let $\mathcal{G} = (\mathcal{V}, \mathcal{E})$ denote a graph with node set $\mathcal{V} = \{v_1, v_2, \cdots, v_N\}$ and edge set $\mathcal{E}$, where $N$ is the number of nodes. The topology of graph $\mathcal{G}$ can be represented by its adjacency matrix $\mathbf{A} = [a_{ij}] \in \{0, 1\}^{N \times N}$, where $a_{ij} = 1$ if and only if there exists an edge $e_{ij} = (v_i, v_j)$ between nodes $v_i$ and $v_j$. The degree matrix $\mathbf{D}$ is a diagonal matrix with diagonal element $d_i = \sum_{i=1}^{N} a_{ij}$ as the degree of node $v_i$. $\mathcal{N}(v_i) = \{v_j | (v_i, v_j) \in \mathcal{E}\}$ stands for the neighbourhoods of node $v_i$.

$\mathbf{X} \in \mathbf{R}^{N \times F}$ and $\mathbf{H} \in \mathbf{R}^{N \times F'}$ denote the collection of node attribute and representation with the $i^{th}$ row, i.e., $\mathbf{x}_i \in \mathbb{R}^F$ and $\mathbf{h}_i \in \mathbb{R}^{F'}$, corresponding to node $v_i$, where $F$ and $F'$ stand for the dimensions of attribute and representation.

**Preliminaries:** Most Graph Neural Networks (GNNs) follow an aggregation-combination strategy [6], where each node representation is iteratively updated by aggregating node representations of neighbourhoods and combining the aggregated representation with the node representation itself as follows

$$\bar{\mathbf{h}}_v^k = \text{AGGREGATE}^k \left( \{ \mathbf{h}_u^{k-1} | u \in \mathcal{N}(v) \} \right), \quad \mathbf{h}_v^k = \text{COMBINATE}^k \left( \mathbf{h}_v^{k-1}, \bar{\mathbf{h}}_v^k \right), \quad (1)$$

where $\bar{\mathbf{h}}_v^k$ stands for the aggregated representation from neighbourhoods. The aggregation operation is the most critical part of the message passing framework, and most GNNs utilize summarization or average function as the implementation of $\text{AGGREGATE}^k$. Therefore, most GNNs can be unified under the following formula

$$\mathbf{h}_v^k = \sigma \left( \sum_{u \in \mathcal{N}(v)} c_{uv}^k \mathbf{h}_u^{k-1} \mathbf{W}^k \right), \quad (2)$$

where $\mathbf{W}^k$ is the learnable parameter and the $\sigma(\cdot)$ is the nonlinear mapping function. The scalar $c_{uv}$ is the averaging weight, which determines the scheme of aggregation. GNNs can be divided into two categories according to the design of $c_{uv}$. The methods in the first category fix $c_{uv}$ by regarding topology information as perfect. For example, GCN [3] and SGC [25] set $c_{uv}^k = 1/(\sqrt{(d_u + 1)(d_v + 1)})$, while GIN [26] sets $c_{uv}^k = 1$ for $u \neq v$ and $c_{vv}^k = 1 + \epsilon^k$. The methods in the second category tend to learn $c_{uv}$ by considering topology as noisy. For example, GPRGNN [27] sets $c_{uv}^k = \gamma^k/(\sqrt{(d_u + 1)(d_v + 1)})$ with $\gamma^k$ as learnable real number. Graph Attention Network (GAT) [16], Gated Attention Network (GaAN) [17] and Probabilistic GCN [18] model the propagation weights as the function of the attributes of connecting nodes via normalized attention mechanism as

$$c_{uv} = softmax(e_{uv}) = \exp(e_{uv}) / \sum_{k \in \mathcal{N}(u)} \exp(e_{uk}). \quad (3)$$

where $e_{uv}$ denotes the similarity between nodes $v$ and $u$. The similarity function of attributes can be specified as $e_{uv}^{GAT} = \text{LeakyReLU}(\mathbf{b}^k[\mathbf{W}\mathbf{h}_u^{k-1}||\mathbf{W}\mathbf{h}_v^{k-1}])$, $e_{uv}^{GaAN} = (\mathbf{W}\mathbf{h}_u^{k-1})^T \mathbf{O}^k (\mathbf{W}\mathbf{h}_v^{k-1})$ and $e_{uv}^{PGCN} = -(\mathbf{W}\mathbf{h}_u^{k-1} - \mathbf{W}\mathbf{h}_v^{k-1})^T \mathbf{\Sigma} (\mathbf{W}\mathbf{h}_u^{k-1} - \mathbf{W}\mathbf{h}_v^{k-1})$ where $\mathbf{b}^k$, $\mathbf{O}^k$ and $\mathbf{\Sigma}^k$ are learnable parameters. Some efforts have been paid to simplify them, such as setting $\mathbf{O}^k$ as identity matrix, i.e., $\mathbf{O}^k = \mathbf{I}$ or constraining $\mathbf{\Sigma}^k$ as diagonal matrix. FAGCN [21] relaxes the non-negative constraint in GAT to real number by employing $tanh(\cdot)$ instead of softmax.

GAT [16] proposes to improve the stability by employing multi-head attentions as in Transformer [28]. The multi-head attention essentially performs multi-channel propagation. DMP [20] formates multi-channel propagation as diverse message passing by enhancing Eq. (2) via feature-wise propagation weights as

$$\mathbf{h}_v^k = \sigma \left( \sum_{u \in \mathcal{N}(v)} \mathbf{c}_{uv}^k \odot \mathbf{h}_u^{k-1} \mathbf{W}^k \right), \quad (4)$$

where $\odot$ denotes the element-wise product of vectors and the learnable propagation weight vector $\mathbf{c}_{vu}^k$ has the same length as the node representation $\mathbf{h}_u^{k-1}$. To reduce the model complexity, DMP presents two schemes to efficiently learn $\mathbf{c}_{vu}^k$'s.

**Analysis:** By analyzing the single-channel propagation in Eq. (4) and the multi-channel propagation in Eq. (4), it can be observed that existing methods possess two serious drawbacks as shown in the blue dashed box in Fig. 1:

- **Propagations to each node are irrelevant.** As shown in Eq. (3), the propagation weight, which is based on the contents of the two connected nodes, models *pairwise* relationship. Thus, the different propagations to each node are irrelevant to each other. However, the node representation, which is obtained from aggregation over neighbourhood, should reflect global characteristics of its neighbourhood, such as high-order relationship.

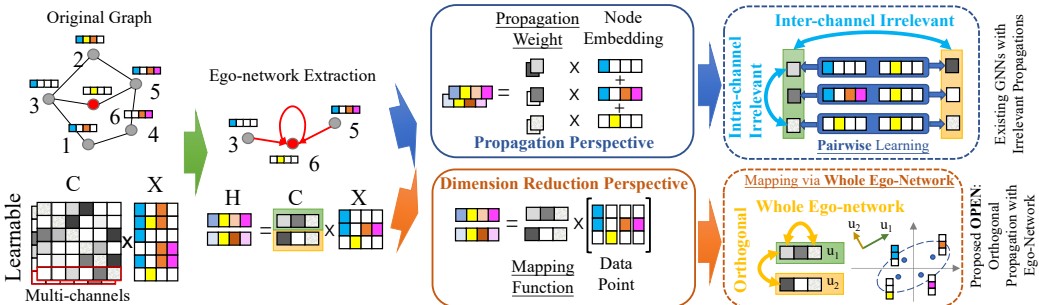

Figure 1: Comparisons between existing GNNs with irrelevant propagations and the proposed OPEN (Orthogonal Propagation with Ego-Network modeling). The essential difference between them is the different perspectives on ego-network modeling. **Upper Part**: Existing GNNs are from the perspective of propagation with pairwise propagation weights learning, and thus both inter-channel and intra-channel propagations are irrelevant. **Lower Part:** The proposed OPEN is from the perspective of dimension reduction of data points, and thus mapping functions are from whole ego-network modeling and orthogonal between different channels.

- **Propagations in multi-channels are irrelevant.** As shown in Eq. (4), propagations in different channels are free to learn without any specific constraints. Thus, the propagation weights in different channels may be similar and redundant. However, to obtain stable and complementary representations from different channels, propagation schemes in different channels should be diverse enough.

In summary, neither the relationship between the propagations to each node nor the relationship between propagations in different channels is considered. Thus, propagation weights are free to be determined by the labelled and the specific task, and thus the models tend to be overfitting [24].

## 3    Methodology

To tackle the issues mentioned above, Orthogonal Propagation with Ego-Network modeling (OPEN) is proposed by modeling these two kinds of relationships between propagations in subsections 3.1 and 3.2, respectively. Section 3.3 provides implementation details. Finally, section 3.4 theoretically analyzes its capability on preventing over-smoothing issue.

### 3.1    Ego-Network Modeling

Firstly, the relationship between propagations to one node is modeled. By ignoring the layer $k$ and considering the first layer, the propagation component in Eq. (2) can be reformulated as

$$\mathbf{h}_v = \sum_{u \in \mathcal{N}(v)} c_{uv} \mathbf{h}_u = \mathbf{c}_v \mathbf{H}_v, \tag{5}$$

where $\mathbf{H}_v \in \mathbb{R}^{|\mathcal{N}_v| \times F}$ stands for the matrix containing the representations of nodes in $\mathcal{N}_v$ and $\mathbf{c}_v \in \mathbb{R}^{|\mathcal{N}_v|}$ represents the propagation weights from nodes in $\mathcal{N}_v$ to node $v$. Eq. (5) can be interpreted from two perspectives as shown in Fig. 1. From the perspective of message passing, the representation of node $v$, i.e., $\mathbf{h}_v$ is the aggregation of embeddings of its neighbourhood, which are the rows of $\mathbf{H}_v$, and propagation weights often model pairwise relationship as shown in the blue solid box in Fig. 1. This perspective induces the propagations to each node irrelevant as shown in the blue dashed box in Fig. 1.

From another perspective, $\mathbf{h}_v \in \mathbb{R}^{1 \times F}$ can be regarded as the one-dimensional representations of $F$ data points, each of which corresponds to a $\mathcal{N}_v$-dimensional column vector of $\mathbf{H}_v$. And, $\mathbf{c}_v \in \mathbb{R}^{|\mathcal{N}_v|}$ can be seen as the mapping function, which reduces the dimension from $\mathbb{R}^{|\mathcal{N}_v|}$ to $\mathbb{R}$ as shown in the orange solid box in Fig. 1. From the perspective of dimension reduction, the $F$ representations in $\mathbf{h}_v$ should preserve the discriminative information in original $|\mathcal{N}_v|$-dimensional space of $\mathbf{H}_v$. Therefore,

learning the propagation weights $\mathbf{c}_v$ in ego-network of node $v$ is converted to the problem of seeking dimension reduction mapping function.

Among the existing dimension reduction methods, principal component analysis (PCA) is the widely-adopted unsupervised one. The principal components correspond to the mapping functions. To facilitate the description, the subscripts of $\mathbf{H}_v$ and $\mathbf{c}_v$ are removed, and $\mathbf{H}$ is represented as the collection of column vectors as $\mathbf{H} = \{\mathbf{h}_{\cdot,1}, \mathbf{h}_{\cdot,2}, ..., \mathbf{h}_{\cdot,F}\}$, each of which is an $|\mathcal{N}_v|$-dimensional vector. By denoting the mean and covariance matrix of data matrix $\mathbf{H}$ as $\bar{\mathbf{h}} = \frac{1}{F} \sum_{j=1}^{F} \mathbf{h}_{\cdot,j}$ and $\mathbf{S} = \frac{1}{F} \sum_{j=1}^{F} (\mathbf{h}_{\cdot,j} - \bar{\mathbf{h}})(\mathbf{h}_{\cdot,j} - \bar{\mathbf{h}})^T$, respectively, the principal components are the eigenvectors of covariance matrix $\mathbf{S}$, i.e.

$$\mathbf{S}\mathbf{u}_j = \lambda_j \mathbf{u}_j, \quad j = 1, 2, ..., |\mathcal{N}_v| \tag{6}$$

where $\mathbf{u}_j$ is the eigenvector corresponding to the eigenvalue $\lambda_j$ as shown in the orange dashed box in Fig. 1. By sorting $\lambda_j$ in descending order, the eigenvector $\mathbf{u}_1$ corresponding to largest eigenvalue $\lambda_1$ is the first principal component. By using the $\mathbf{u}_1$ as the mapping function $\mathbf{c}$, the obtained 1-dimensional representation possesses the largest variance, and thus preserves the discriminative ability.

**Ego-network of Hub Nodes:** The computational complexity, which consists of construction of $\mathbf{S}$ and EVD, is $\mathcal{O}(|\mathcal{N}_v|^2 F)$. Thus, it is efficient for most nodes with small neigbourhood. However, it may be inefficient for nodes with large neighbourhood, such as hub nodes. Thus, we present the treatment for them. By denoting $\tilde{\mathbf{H}} = \{\mathbf{h}_{\cdot,1} - \bar{\mathbf{h}}, \mathbf{h}_{\cdot,2} - \bar{\mathbf{h}}, ..., \mathbf{h}_{\cdot,F} - \bar{\mathbf{h}}\}$, Eq (6) can be written as $\frac{1}{N}\tilde{\mathbf{H}}\tilde{\mathbf{H}}^T\mathbf{u}_j = \lambda_j\mathbf{u}_j$. By multiplying both sides by $\tilde{\mathbf{H}}^T$, it holds

$$\frac{1}{N}\tilde{\mathbf{H}}^T\tilde{\mathbf{H}}\tilde{\mathbf{H}}^T\mathbf{u}_j = \lambda_j\tilde{\mathbf{H}}^T\mathbf{u}_j. \tag{7}$$

By denoting $\mathbf{z}_j = \tilde{\mathbf{H}}^T\mathbf{u}_j$, Eq. (7) and $\mathbf{u}_j$ can be respectively reformulated as

$$\left(\frac{1}{N}\tilde{\mathbf{H}}^T\tilde{\mathbf{H}}\right)\mathbf{z}_j = \lambda_j\mathbf{z}_j, \quad \mathbf{u}_j = \frac{1}{(F\lambda_j)^2}\tilde{\mathbf{H}}\mathbf{z}_j. \tag{8}$$

Since $\frac{1}{N}\tilde{\mathbf{H}}^T\tilde{\mathbf{H}} \in \mathbb{R}^{F \times F}$, its construction and EVD are efficient. Thus, the complexity of ego-network modeling for hub nodes, which possess a large number of neighbourhoods, i.e., $|\mathcal{N}_v|$, is reduced from $\mathcal{O}(|\mathcal{N}_v|^2 F)$ to $\mathcal{O}(|\mathcal{N}_v|F^2)$. Therefore, ego-network for hub node can be efficiently modeled.

This new perspective and corresponding ego-network modeling provide following three attractive characteristics:

- **Beyond pairwise modeling:** Since the covariance matrix $\mathbf{S} \in \mathbb{R}^{|\mathcal{N}_v| \times |\mathcal{N}_v|}$ essentially models the correlation between nodes in the ego-network $\mathcal{N}_v$, its eigenvector represents the high-order characteristics of ego-network, which is beyond the pairwise relationship between two connected nodes. Therefore, the propagations to each node are jointly modeled.

- **Preventing overfitting issue:** The mapping function, i.e., the propagation weights, is inferred from the ego-network instead of learning from labelled nodes for specific task. The model parameters of propagation weight learning are omitted, and thus it prevent overfitting.

- **Hight efficiency:** The eigenvalue decomposition (EVD) is performed over the ego-networks whose sizes are much smaller than that of whole network, and thus the EVD is efficient.

### 3.2 Orthogonal Propagations in Multi-channels

Since multi-channel introduces multiple groups of trainable parameters, it tends to be more serious overfitting than the single-channel case. To tackle this issue and enhance the robustness, an intuitive strategy is to boost the diversities of different channels to provide complementary effects. GNNs may implement this requirement by designing diverse propagation scheme. However, it is difficult to constrain pairwise weight learning to be diverse in an efficient manner, since diversity often requires to impose computationally expensive orthogonality constraints.

Fortunately, the employed PCA in ego-network modeling in section 3.1 simultaneously produces $|\mathcal{N}_v|$ orthogonal components $\{\mathbf{u}_1, \mathbf{u}_2, ..., \mathbf{u}_{|\mathcal{N}_v|}\}$, which corresponds to different mapping functions. Note that the PCA is efficient on ego-networks with a small number of nodes. Thus, these $|\mathcal{N}_v|$

orthogonal components $\{\mathbf{u}_1, \mathbf{u}_2, ..., \mathbf{u}_{|\mathcal{N}_v|}\}$ can be employed as the propagation weights in $|\mathcal{N}_v|$ different channels to realize the diversity requirement. By letting $\mathbf{c}_v^j = \mathbf{u}_j$ as the propagation weights over the ego-network of node $v$ on $j^{th}$ channel, Eq. (5) can be extended to multi-channel form as

$$\mathbf{h}_v^j = \mathbf{c}_v^j \mathbf{H}_v, \quad j = 1, 2, ..., |\mathcal{N}_v| \tag{9}$$

**Discussion:** Ortho-GConv [29] also introduces the orthogonality in GNNs. However, it is very different from the proposed OPEN on both motivation and methodology. From the perspective of motivation, Ortho-GConv tends to alleviate oversmoothing issue, while OPEN aims at overfitting issue. From the perspective of methodology, Ortho-GConv imposes orthogonality on feature transformation $\mathbf{W}^k$, while OPEN presents orthogonal propagation, which is more challenging and essential in GNNs.

### 3.3 Model Details

Subsections 3.1 and 3.2 provide the basic ideas of ego-network modeling and extension to multi-channels with orthogonal propagations. This section presents the details on implementation.

Firstly, the number of channels is set as $J$ in advance. Since the number of principal components for each ego-network is the same as the number of neighbourhood of the center node, i.e., $|\mathcal{N}|_v$, these numbers varies over nodes. To make representations of nodes comparable, the number of channels is set as a hyper-parameter. For the nodes, whose number of neighboood is less than $J$, the lacking components are set as zero vector, i.e., $\mathbf{c}_v^j = \mathbf{0}$ for $j = |\mathcal{N}|_v + 1, ..., J$. The tuning experiments on $J$ are shown in subsection 4.5.

Secondly, the final representation of node can be obtained by combing representations in multi-channels from Eq. (9), such as summarization or concatenation. To boost the expressive capability, attention mechanism is employed to combine different representations with shared learnable parameter $\mathbf{b}$, i.e.

$$\mathbf{h}_v = \sum_{j=1}^{J} \alpha_v^j \mathbf{h}_v^j, \quad \alpha_v^j = \frac{\exp(\mathbf{b}^T \mathbf{h}_v^j)}{\sum_{i=1}^{J} \exp(\mathbf{b}^T \mathbf{h}_v^i)}. \tag{10}$$

Similar to existing GNNs, OPEN is trained by feeding the final representation $\mathbf{h}_v$'s into the cross-entropy between predicted and ground-truth labels on labelled nodes for semi-supervised task.

Thirdly, propagation weights inferred from the original node attributes are fixed for all layers. Although it can obtain optimal solution by inferring propagation weights for the next layer from the representations obtained in last layer, the inference process and the back-propagation process can't be seamlessly combined. For the consideration of computational complexity, propagation weights inferred from the original node attributes are employed for all layers.

### 3.4 Theory Analysis on Preventing Over-smoothing

Over-smoothing issue [10, 11], i.e., representations of all nodes converge to points independent from their original attributes, has been identified as the most serious issue to prevent GNNs from being deep. The over-smoothing also leads to the exponential loss of expressive power for node classification [12]. According to [11], over-smoothing issues is caused by that the eigenvector corresponding to the largest eigenvalue of normalized adjacency matrix ($\lambda_1 = 1$) is determined by degree, i.e., $\mathbf{u}_1 = \mathbf{D}\mathbf{e}$ for asymmetric normalization or $\mathbf{u}_1 = \mathbf{D}^{\frac{1}{2}}\mathbf{e}$ for symmetric normalization, where $\mathbf{e}$ is the vector of ones. Thus, the node representations after infinite layers are

$$\mathbf{H}^\infty = \lim_{l \to \infty} \tilde{\mathbf{A}}^l \mathbf{X} = \mathbf{u}_1 \mathbf{u}_1^T \mathbf{X} = \mathbf{u}_1 \left( \mathbf{u}_1^T \mathbf{X} \right) = \mathbf{u}_1 \mathbf{w}^T, \tag{11}$$

where $\mathbf{w} = \mathbf{X}^T \mathbf{u}_1$ is the weighted combination of attributes from all nodes, and thus it is shared by all nodes. Therefore, the final representation $\mathbf{H}^\infty$ is determined by node degree $\mathbf{D}$. To alleviate this issue, many efforts have been paid. Some of them modify the topology, such as DropEdge [30] and GRAND [31], while others revise the outputs from layers, such as PairNorm [32] and Inflation [33].

Different from existing methods, the proposed OPEN essentially refines the topology $\mathbf{A}$ according to the attributes of nodes in ego-network $\mathbf{X}_v$, i.e., $\hat{\mathbf{A}} = f(\mathbf{A}, \mathbf{X})$. Therefore, it is intuitive that the $\hat{\mathbf{u}}_1$ corresponding $\hat{\mathbf{A}}$ is the composition of $\mathbf{D}$ and $\mathbf{X}$. According to Eq. (11), the final representation is relevant to $\mathbf{X}$, i.e. preventing over-smoothing issue. The following theorem is proved in Appendix.

Table 1: Statistics of datasets used for node-level tasks.

| Dataset | #Nodes | #Edges | #Features | #Classes |
|---|---|---|---|---|
| Cora | 2,708 | 5,429 | 1,433 | 7 |
| Citeseer | 3,327 | 4,732 | 3,703 | 6 |
| Pubmed | 19,717 | 44,338 | 500 | 3 |
| Amazon-Computers | 13,752 | 245,861 | 767 | 10 |
| Amazon-Photo | 7,650 | 119,081 | 745 | 8 |
| Coauthor-CS | 18,333 | 81,894 | 6,805 | 15 |
| Coauthor-Physics | 34,493 | 247,962 | 8,415 | 5 |

**Theorem 1.** *The representation of each node from OPEN is relevant to the principal components of its corresponding ego-network's attribute.*

# 4   Evaluations

Firstly, this section provides experimental setups, including dataset, baselines and implementation details. Then, the node classification results are analyzed followed by hyper-parameter tuning. Finally, the capabilities on preventing over-smoothing and overfitting are verified.

## 4.1   Datasets

To comprehensively evaluate the proposed OPEN, 7 widely used datasets are employed. Statistics of datasets are shown in Table 1. These datasets can be divided into three categories.

- **Citation Networks.** Cora, Citeseer, and Pubmed, which are widely used to verify GNNs, are standard citation network benchmark datasets [34, 35]. In these networks, nodes and edge represent papers and citations between them, respectively. Words in the paper are employed to represent the node feature in bag-of-word form. The academic topic of paper is taken as the label of node.

- **Co-purchase Networks.** Amazon-Computers (Computers) and Amazon-Photo (Photo) are two networks of co-purchase relationships [36]. In these networks nodes represent goods and edges stand for the connected two goods being frequently bought together. Each node owns a bag-of-words feature extracted from product reviews. The categories of the goods are employed as the label of node.

- **Coauthor Networks.** Coauthor-CS (CS) and Coauthor-Physics (Physics) are two co-author networks based on the Microsoft Academic Graph from the KDD Cup 2016 challenge [36]. In these networks nodes represent authors and edges stand for that the connected two authors have co-authored a paper. Each node owns a bag-of-words feature based on paper keywords. The most active research fields of the authors are employed as the label.

## 4.2   Baselines

To verify the superiority of the proposed OPEN, 15 baseline methods are employed for performance comparison. These methods are divided into two categories. The first category consists of basic methods for graph data, including the multiple layer perception (MLP), Logistic Regression (LogReg) and Label Propagation (LP) [37], Chebyshev [4], Graph Convolutional Network (GCN) [3], Graph Attention Network (GAT) [16], GraphSAGE [5], Simple Graph Convolution (SGC) [25], MoNet [38]. The methods in the second category possess some attractive characteristics, such as preventing over-smoothing issue and high accuracy, etc This category contains Personalized Propagation of Neural Predictions (APPNP) [39], Jumping Knowledge Networks (JKNet) [40], GCN with Initial residual and Identity mapping (GCNII) [41], Generalized PageRank GNN (GPRGNN) [27], Diverse Message Passing (DMP) [20] and Constrained GAT (C-GAT) [24]. We employ the authors' implementations for all baseline methods with the default hyper-parameters.

Table 2: Node classification performance in terms of micro-f1 scores.

| dataset | Cora | Pubmed | Citeseer | Computer | Photo | CS | Physics |
|---|---|---|---|---|---|---|---|
| MLP | 74.82±2.22 | 63.76±0.78 | 74.05±2.10 | 70.48±0.28 | 78.69±0.30 | 88.30±0.70 | 88.90±1.10 |
| LogReg | 70.10±2.30 | 61.00±2.20 | 71.10±3.10 | 76.80±5.70 | 79.20±6.50 | 86.40±0.90 | 86.70±1.50 |
| LP | 78.00±0.20 | 75.30±0.20 | 69.00±0.50 | 70.80±0.00 | 67.80±0.00 | 74.30±0.00 | 90.20±0.50 |
| Chebyshev | 82.20±0.50 | 81.80±0.50 | 73.40±0.30 | 72.60±0.00 | 84.30±0.00 | 91.50±0.00 | 92.10±0.30 |
| MoNet | 82.30±1.30 | 80.20±2.00 | 74.60±2.30 | 88.60±2.20 | 91.20±1.30 | 90.80±0.60 | 92.50±0.90 |
| GCN | 85.77±0.25 | 88.13±0.28 | 73.68±0.31 | 86.51±0.54 | 92.42±0.22 | 94.55±0.24 | 95.58±0.20 |
| GAT | 87.37±0.30 | 87.62±0.26 | 74.32±0.27 | 86.93±0.39 | 92.56±0.35 | 93.98±0.22 | 95.63±0.18 |
| GraphSAGE | 87.77±1.04 | 88.42±0.50 | 71.09±1.30 | 83.11±0.23 | 90.51±0.25 | OOM | 95.52±0.54 |
| SGC | 86.20±0.12 | 87.50±0.25 | 78.10±0.13 | 80.09±0.16 | 88.25±0.52 | 89.62±0.42 | 90.05±0.50 |
| GCNII | 88.49±2.78 | 89.57±1.56 | 77.08±1.21 | 86.13±0.51 | 90.98±0.93 | 93.88±0.40 | 96.02±0.15 |
| APPNP | 87.87±0.85 | 89.40±0.61 | 76.53±1.33 | 81.99±0.26 | 91.11±0.26 | 94.92±0.20 | 95.84±0.22 |
| JKNet | 88.93±1.35 | 87.68±0.30 | 74.37±1.53 | 77.80±0.97 | 87.70±0.70 | 92.32±0.72 | 95.71±0.21 |
| C-GAT | 88.40±0.30 | 87.60±0.30 | 79.90±0.30 | OOM | OOM | OOM | OOM |
| GPRGNN | 88.37±1.33 | 89.05±0.52 | 78.88±1.70 | 89.43±0.86 | 94.34±0.35 | 94.76±0.20 | 96.39±0.20 |
| DMP | 86.68±1.13 | 89.67±0.58 | 76.48±1.73 | 87.62±0.81 | 94.06±0.45 | **95.02±0.29** | 94.74±0.20 |
| OPEN | **89.31±0.53** | **90.05±0.23** | **80.42±0.72** | **90.93±0.29** | **95.43±0.27** | 94.99±0.13 | **96.92±0.05** |

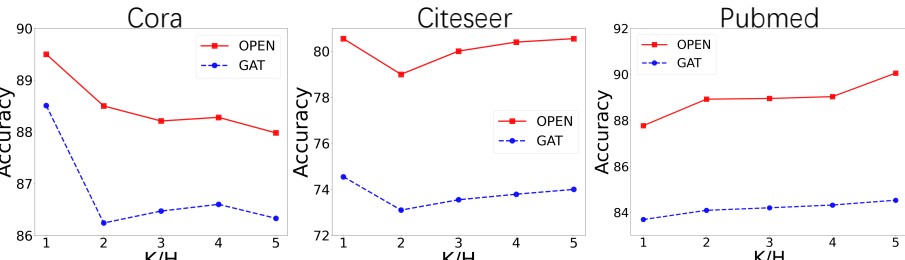

Figure 2: The impact of the number of channels on the performance on Cora, Citeseer and Pubmed.

## 4.3  Implementation Details

The proposed OPEN employs 2-layers network with $K = 5$ channels except for the hyper-parameter tuning (Sec. 4.5) and over-smoothing investigation (Sct 4.6). The whole network is trained in an end-to-end manner using the Adam optimizer with an initial learning rate of 0.001. The maximum number of epochs is set up to 1000. Besides, early stopping with a patience of 50 is also utilized. For all datasets, we randomly split nodes of each class in to 60%, 20% and 20% for training validation and testing, and run on test sets over 10 random splits, as suggested in [42]. For fair comparison, the performance of all the methods, including baseline methods, are obtained on the same splits.

## 4.4  Result Analysis

The node classification results are shown in Tab. 2. The proposed OPEN achieves the new SOTA on 6 networks in all 7 networks. Note that the proposed OPEN not only outperforms all basic GNNs, such as GCN and GraphSAGE, but also significantly beats almost all SOTA methods, such as GCNII, GPRGNN and DMP. These demonstrate the superiority of the proposed OPEN on classification accuracy. This can be attributed to both the effectiveness of ego-network modeling on capturing high-order semantic information and the high diversity representations obtained from orthogonal propagation in multi-channels. Besides, the proposed OPEN also outperforms other extensions to GAT, such as C-GAT and DMP. This indicates that compared to other enhancements to the propagation weights learning and multi-channel learning, ego-network modeling and orthogonal propagation are more effective. In summary, these effectivenesses show that the modeling the irrelevance between propagations is important and necessary for GNNs.

## 4.5  Hyper-parameter Tuning and Ablation Study

The only hyper-parameter in the proposed OPEN is the number of principal components, $K$, which is also corresponds to the number of channels. This section investigates the impact of $K$ on classification

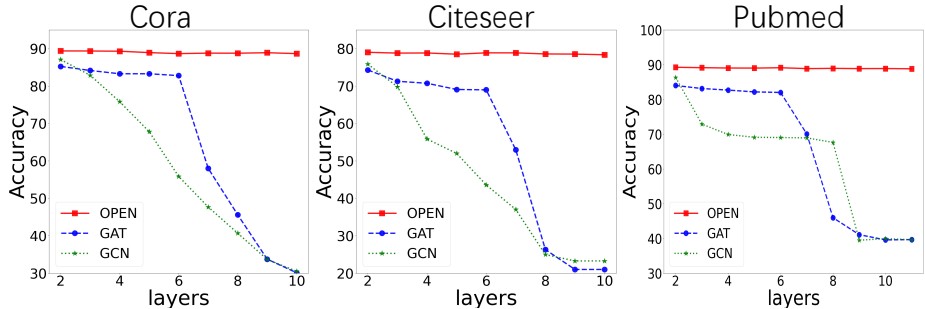

Figure 3: Node classification results with various model depths on Cora, Citeseer and Pubmed.

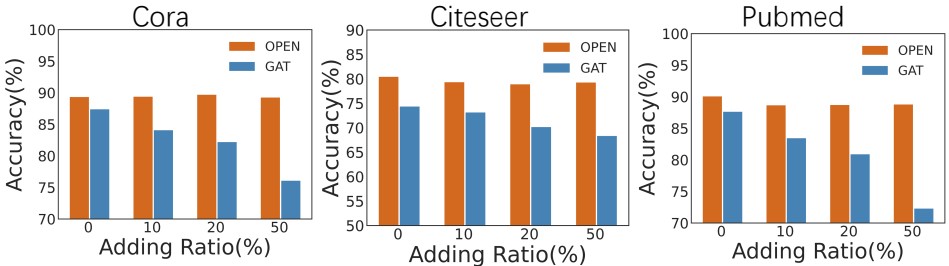

Figure 4: Node classification performance on graphs with randomly adding noisy edges.

accuracy. To exclude the influences from different datasets, the impact of the number of channels in GAT, i.e., the number of attention heads, is also investigated. Fig.2 shows that OPEN consistently outperforms GAT on all the number of channels. However, the trends with different number of channels are different on different datasets. In most cases, the superiority of OPEN is more remarkable with large $K$ compared to GAT. Thus, the $K$ is set as 5 for other experiments by balancing the accuracy and running time.

**Ablation Study:** The hyper-parameter tuning experiments also enable the ablation study. 1) When $K = 1$, only the ego-network modeling component works. The significant outperformance compared to GAT demonstrates that the relevant propagation in OPEN is more superior than irrelevant propagations in GAT. 2) The consistent outperformances on all $K$'s reveal the importance of orthogonal propagations in multi-channels, which are also relevant. The ablation study illustrates that both relevance modelings are critical and necessary.

### 4.6 Preventing Over-smoothing Issue

Section 3.4 theoretically analyzes the capability of OPEN on preventing oversmoothing issue. This section provides experimental evaluations. The node classification performance changes of GCN, GAT and OPEN with various model depths are shown in Fig. 3. GCN tends to be over-smoothing via only a few layers, since it employs topology-induced single-channel propagation. GAT alleviates the overmoothing issue by introducing irrelevant multi-channel propagations. Unfortunately, it also be oversmoothing after few layers. By enhancing the GAT with relevant multi-channel propagations, OPEN significantly prevents the over-smoothing issue. After many layers, the performances of OPEN do not remarkably drop. This matches the theoretical results in Section 3.4, and verifies the capability of relevant propagation on preventing oversmoothing issue.

### 4.7 Robustness

This section investigates the robustness of the proposed OPEN, which indicates whether the OPEN is able to overcome the overfitting problem, as in [24]. To this end, experiments are conducted by randomly perturbing the edges in the testing graph. Specifically, a set of nodes are randomly selected according to a given sampling ratio, and then one edge is randomly added on these nodes.

Fig. 4 reports the classification performance on testing graphs with different ratios of nodes. The performances of GAT significantly drop as the ratios of nodes increase, which indicate it tends to be overfitting to the given data. In contrary, the performance drops of the proposed OPEN are very slight. It indicates the robustness of OPEN. This robustness may be attributed to two aspects. Firstly, the propagation weights of OPEN are inferred from neighbourhoods instead of learning from labelled data. Secondly, the orthogonal propagations in multi-channels promote the diversity, and thus enhance the robustness. This investigation shows that relevant propagation modeling can promote the robustness of the GNNs.

## 5    Conclusions

This paper identifies the *irrelevant propagations* issue in Graph Neural Networks (GNNs), which makes models exposed to overfitting to the labelled data. The propagation irrelevance includes 1) the propagations to one node are irrelevant, 2) propagations in multi-channels are irrelevant. This paper presents a novel Orthogonal Propagation with Ego-Network modeling (OPEN) to model these two kinds of relevances between propagations. By interpreting the propagations to one node from the perspective of dimension reduction, propagation weights are inferred from the principal components of ego-network, which are orthogonal to each other. Theoretical analysis and experimental evaluations reveal four attractive characteristics of the proposed OPEN as modeling high-order relationship beyond pairwise one, preventing overfitting issue, robustness, and high efficiency. These attractive characteristics reveal the importance of modeling propagation relevance.

## Acknowledgments and Disclosure of Funding

This work was supported in part by the National Science Fund for Distinguished Young Scholars under Grant 62025602, in part by the National Natural Science Foundation of China under Grant 61972442, Grant 62102413, Grant U1936210, Grants U1803263, Grant 11931015 and Grant 62276187, in part by the Key Research and Development Project of Hebei Province of China under Grant 20350802D and 20310802D; in part by the Natural Science Foundation of Hebei Province of China under Grant F2020202040, in part by the Natural Science Foundation of Tianjin of China under Grant 20JCYBJC00650, in part by the China Postdoctoral Science Foundation under Grant 2021M703472, in part by the XPLORER PRIZE, and in part by the Fundamental Research Funds for Central Universities.

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
