# OpenReview forum: "OPEN: Orthogonal Propagation with Ego-Network Modeling"
_NeurIPS.cc/2022/Conference — NeurIPS 2022 Accept_

### Official Review · Reviewer_aMxP · 2022-07-12

**Rating:** 7
**Confidence:** 3
**Soundness:** 3 good
**Presentation:** 3 good
**Contribution:** 3 good

**Summary:**

In this paper, the authors propose a new GNN model (OPEN) to tackle the irrelevant propagations issues.
The authors first made observations on two limitations of traditional GNNS:
 * The propagation weights to each node are computed in a pairwise fashion, which ignores certain global information for that node's neighborhood.
* There are no constraint on the channel specific propagation weights, which could potentially lead to overfitting.

In OPEN, the propagation weights to a node is computed by eigenvalue decomposition (EVD) of the covariance matrix of node representations in ego-network. It is claimed that this would encode global information from ego-network and prevent overfitting is supervised learning.

Since most of the ego-networks are much smaller in size, the EVD can be computed efficiently (with special treatments for Hub nodes).

Also, by design, the propagation weights from different channels are orthogonal in OPEN, therefore it encourages diversity and reduces overfitting risk.

Empirical results show that the proposed architecture outperforms other GNNs on several node classification tasks.

**Questions:**


* Can the authors provide more context on how the propagation weights are incorporated in back-propagation?
* Why the inference process and the back propagation process can't "be seamlessly combined" for layer specific propagation weights?
* I'd suggest to re-phrase the narrates in Theorem 1 to be more specific. Even over-smoothed representations are "relevant to node attributes", just not related to the node itself.

**Limitations:**

The authors discussed about connections and differences between OPEN and other GNNs with orthogonal constraint. Limitations such as inability to use layer specific propagation weights are discussed.

To my knowledge the work poses no potential negative societal impact.

**Strengths And Weaknesses:**

This paper is well presented and easy to follow. The authors first unifies the expression for different GNNs and formulate the problem as finding the right propagation weights. Then they introduce their proposed method via the dimension reduction point of view.

The empirical results shows the proposed model outperforms other baseline GNNs on various node classification tasks. Ablation study also shows that it can indeed prevent over-smoothing issue.

Even though the method itself is not complicated, I think this could give us a good insight on the limitations of existing GNN architectures.

---

> ### Author Response · Authors · 2022-08-02
> **Response to Reviewer aMxP**
>
> > Q1. Can the authors provide more context on how the propagation weights are incorporated in back-propagation?
>
> R1. The incorporation strategy of OPEN is similar to that of classic GCN. As explained in the Q2, it is inefficient to combine inference process for layer specific propagation weights with the back propagation. Thus, propagation weights inferred from original node attributes in the first layer are employed for all layers. This process is elaborated as follows. Firstly, the propagation weights in each ego-network are obtained from node attributes via PCA. Then, the propagation matrix $\tilde{A}$ is constructed using the obtained propagation weights. Note that the topology structure is not changed, while only the weights are changed. Finally, the constructed propagation matrix is employed for all layers, and the other parameters, including the mapping function $W$ and the weight $b$ for channel combination, are learned via back-propagation. The performance of fixed propagation weights for all layers is compared to those of learning specific weights for each layer. The results are show as follows. Although the performance is slightly improved, its running time significantly increases as shown in the Table R2.
>
> Table R1. Performance Comparisons.
> |Method|Cora|Pubmed|Citeseer| Comput| Photo| CS| Physics|
> |------|---:|-----:|-----:|-----:|-----:|-----:|-----:|
> |GCN    		|85.77  | 88.13 |  73.68 | 86.51  | 92.42 | 94.55  | 95.58|
> |OPEN   		|89.31  | 90.05  | 80.42 | 90.93  | 95.43 |  94.99 | 96.92|
> |OPEN-layer 	|89.39  | 90.21  | 80.55 | 90.76  | 95.61 |  94.99 | 96.98|
>
>
> > Q2.  Why the inference process and the back propagation process can't "be seamlessly combined" for layer specific propagation weights?
>
> R2. This is essentially owed to the reason that the EVD in PCA can’t be efficiently implemented via neural network.
>
> Firstly, the proposed OPEN seeks propagation weights by applying PCA on each ego-network, whose key operation is the EVD of covariance matrix. To make EVD efficient, many approximation algorithms have been presented by leveraging matrix factorization technique. However, it is not trivial to implement them via neural network.
>
> Secondly, OPEN needs O(N) EVD for propagation weights learning in each layer of feedforward step, where N is the number of nodes. The O(N) EVD can’t be efficiently parallelized via GPU for heavy I/O operations. Therefore, the inference process for layer specific propagation weights, which is based on PCA, can ’t be seamlessly combined with neural network, whose optimization is based on back-propagation. To tackle this computational issue, propagation weights inferred from original node attributes in the first layer are employed for all layers. The running time comparisons are shown in Table R2, where OPEN-layer stands for the OPEN with layer specific weights. Compared to the performance improved shown in Table R1？, the computation cost is too high.
>
> Table R2. Running Time Comparison  (in seconds).
> |Method|Cora|Pubmed|Citeseer| Comput| Photo| CS| Physics|
> |------|---:|-----:|-----:|-----:|-----:|-----:|-----:|
> |GCN	|	9.89	|6.23	|5.32	|16.8	|6.59	|19.2	|21.58|
> |GAT	|	10.45	|49.31	|12.85	|95.23	|42.11	|106.06	|201.79|
> |OPEN	|	12.72	|45.02	|15.85	|83.82	|44.84	|107.2	|202.31|
> |OPEN-layer |	271.11	|929.09	|317.8	|1911.36	|957.62	|1927.81	|5443.37|
>
>
> > Q3.  I'd suggest to re-phrase the narrates in Theorem 1 to be more specific. Even over-smoothed representations are "relevant to node attributes", just not related to the node itself.
>
> R3. According to your suggestion, Theorem 1 is re-phrase as:
>
> Theorem 1. The representation of each node from OPEN is relevant to the principal components of its corresponding ego-network’s attribute.

---

### Official Review · Reviewer_NBUs · 2022-07-15

**Rating:** 4
**Confidence:** 3
**Soundness:** 2 fair
**Presentation:** 2 fair
**Contribution:** 2 fair

**Summary:**


This submission studies local refinement in GNNs via orthogonal propagation with ego-net modeling. The proposed methods are designed to address the problem of irrelevant propagation.  Specifically, the relevance between propagations to one node is modeled by whole ego-network modeling and eigenvector decomposition. Meanwhile, the propagations in multi-channels are implemented by employing all principal components, each of which corresponds to one channel. Experimental results on multiple real-world datasets were provided.

**Questions:**

1. it is claimed that the algorithm has high efficiency because evd with respect to the neighborhood is less expensive than evd of the entire network. However, the original propagation in GNN does not require any evd, and such evd needs to be computed for egonets of each node for OPEN. Hence, the efficiency claim is not justified. More explanation or experiments in terms of run time need to be provided

2. the algorithm flow is hard to follow. An algorithm table would be helpful to improve clarify

3. The difference between this work and Ortho-GConv [26] is not clear. In the discussion it is mentioned that "Ortho-GConv tends to alleviate oversmoothing issue, while OPEN aims at overfitting issue." but the entire paper as well as the theoretical analysis shows OPEN sovles overfitting issue. This needs to be clarified.

4. why orthogonal propagation is more challenging than orthorgonizing features as claimed? It seems to the reviewer they are similar.



**Limitations:**

societal impact not discussed

**Strengths And Weaknesses:**

Strength:
1. the design motivation of viewing propagation as dimensionality reduction is new and very interesting.
2. experiments on real datasets show the effectiveness of the algorithms compared with multiple existing works

Weakness:
1. the clarify and organization of the paper can be improved
2. there are typos and grammar mistakes in the paper.
3. the discussion of novelty compared with previous works is not clear

---

> ### Author Response · Authors · 2022-08-02
> **Response to Reviewer NBUs (Part2)**
>
> > Q3. The difference between this work and Ortho-GConv [26] is not clear. In the discussion it is mentioned that "Ortho-GConv tends to alleviate oversmoothing issue, while OPEN aims at overfitting issue." but the entire paper as well as the theoretical analysis shows OPEN sovles overfitting issue. This needs to be clarified.
>
> R3. There may be some misunderstandings. OPEN, which is designed to prevent overfitting issue in irrelevant propagation, possesses the characteristic of preventing over-smoothing issue.
>
> Firstly, OPEN is designed to prevent overfitting issue and is experimental verified. As shown in Constrained GAT (C-GAT) [21], irrelevant propagation makes propagation weights free to be determined by the labelled data for the specific task, thus results GNNs being susceptible to overfitting. The ego-network modeling in OPEN attempts to prevent the overfitting issue by reducing the model complexity. To this end, the learnable propagation weights in irrelevant propagation model are converted as inference from unlabeled graph data via principal component analysis. The ability of OPEN on preventing the overfitting issue is experimentally verified in Section 4.7. As suggested in C-GAT [21], its robustness to noises demonstrates its ability on preventing overfitting. Compared to irrelevant propagation model GAT, the proposed relevant propagation model OPEN is much robust to noises as shown in Figure 4.
>
> Secondly, OPEN possesses the characteristic of preventing the oversmoothing issue. This attractive characteristic is both verified in the view of theory (Theorem 1 in Section 3.4) and experimental evaluation (Figure 3 in Section 4.6).
>
> The difference between OPEN and Ortho-GConv [26] are two-fold.
> From the motivation perspective: Ortho-GConv tends to alleviate the oversmoothing issue, while OPEN aims at the overfitting issue.
> From methodology perspective: Ortho-GConv imposes the orthogonality on feature transformation $W$, while OPEN presents the orthogonal propagation.
>
>
> >Q4. why orthogonal propagation is more challenging than organizing features as claimed? It seems to the reviewer they are similar.
>
> R4. Compared to the orthogonal feature transformation presented in Ortho-GConv, challenges of orthogonal propagation are two-fold.
>
> The first challenge is how to reasonably define orthogonal propagation. Orthogonality is a concept of linear algebra, and orthogonal matrix is a matrix $X$, which follows $X^TX = I$, where $I$ is the identity matrix. Thus, it is direct to employ this definition to the feature transformation matrix $X$ in Ortho-GConv. However, it is not trivial to define orthogonal propagation. Since the propagation matrix $A$ is a given adjacency matrix, making it follow the definition of the orthogonal matrix is unconscionable. To overcome this difficulty, OPEN assigns the orthogonality constrains to the propagation weights of one node in two channels.
>
> The second challenge is how to elegantly obtain orthogonal propagation weights for each node. To this end, OPEN proposes to perform PCA on each ego-network, since the mapping directions in PCA are from EVD, where the eigenvectors are orthogonal.
>
> Therefore, orthogonal propagation is challenging.

---

> ### Author Response · Authors · 2022-08-02
> **Response to Reviewer NBUs**
>
> > Q1. it is claimed that the algorithm has high efficiency because evd with respect to the neighborhood is less expensive than evd of the entire network. However, the original propagation in GNN does not require any evd, and such evd needs to be computed for egonets of each node for OPEN. Hence, the efficiency claim is not justified. More explanation or experiments in terms of run time need to be provided
>
> R1. There may be some misunderstandings.
>
> Firstly, we only claim that “the EVD is efficient. ” (Line 176-177) and “the ego-network modeling is highly efficient” (Line 167 -177) in the paper instead of “algorithm has high efficiency” in the review comment. As discussed between Lines 158 -166, the complexity of EVD on one ego-network is $\{O}( |\mathcal{N}_v|F^2)$, and those on all ego-networks is $\{O}( |\mathcal{E}|F^2)$, where $|\mathcal{E}|$ is the number of edges in the graph. Thus, we justify the efficiency of EVD and ego-network modeling.
>
> Secondly, the proposed OPEN is as efficient as GAT. Each message passing step of OPEN has the same complexity as that of GAT, i.e., $\{O}( |\mathcal{E}|F^2)$. The whole OPEN consists of ego-network modeling and message passing step, whose complexities are both $\{O}( |\mathcal{E}|F^2)$. Thus, the total complexity of OPEN is $\{O}( |\mathcal{E}|F^2)$, and as the same as that of GAT. The running time comparisons is shown in Table R1, where OPEN-W and OPEN-P represents the time for weight calculation and propagation, respectively. The running time of GAT and OPEN is similar. Note that the running time of GAT and OPEN is longer than that of GCN, due to their multiple-channel propagations and combinations. These additional experiments and discussions are added to Section B.4 of the Appendix.
>
> Table R1. Running time in seconds.
> |Method|Cora|Pubmed|Citeseer| Comput| Photo| CS| Physics|
> |------|---:|-----:|-----:|-----:|-----:|-----:|-----:|
> |GCN|9.89|6.23|5.32|16.8|6.59|19.2|21.58|
> |GAT|10.45|49.31|12.85|95.23|42.11|106.06|201.79|
> |OPEN-W|2.61|8.93|3.05|18.46|9.22|18.39|52.94|
> |OPEN-P|10.11|36.09|12.8|65.36|35.62|88.81|149.37|
> |OPEN|12.72|45.02|15.85|83.82|44.84|107.2|202.31|
>
>
>
> In summary, the EVD and ego-network modeling is highly efficient and the proposed OPEN is as efficient as vanilla GAT.
>
> > Q2.  the algorithm flow is hard to follow. An algorithm table would be helpful to improve clarify
>
> R2. Thanks for your suggestion. We provide two algorithms in the Section C of Appendix.

---

> ### Comment · Reviewer_NBUs · 2022-08-08
> **response to rebuttal**
>
> The reviewer read the response. And found the concerns to fully addressed.
>
> For  Q1  " The proposed OPEN  possesses some attractive characteristics: modeling high-order relationships beyond pairwise one, robustness, and high efficiency. " (line 60-61) in the submission, the authors indeed claim OPEN is with high efficiency instead of evd. From the provided experiment in the rebuttal, the reviewer failed to see this.
>
> For Q3 . In the discussion it is mentioned that "Ortho-GConv tends to alleviate oversmoothing issue, while OPEN aims at overfitting issue." but the entire paper, as well as the theoretical analysis, shows OPEN sovles [[oversmoothing]] issue (there was a typo in my original comments)
>
> Hence, the reviewer still finds they solve the same issue, and was not fully convinced of the novelty compared with Orth-GCov.

---

> > ### Author Response · Authors · 2022-08-09
> > **Response to Reviewer NBUs (Part3)**
> >
> > Dear Reviewer NBUs
> >
> > Thanks for your additional comments. We would like to express our sincere appreciation to you for your insightful comments and compliments to our paper. We hope that you can kindly consider our responses in making the final decision.
> >
> > > Q1+. Algorithm Efficiency.
> >
> > R1+. To demonstrate the efficiency, we compare OPEN with other GNNs, which model high-order relationships, in terms of computational complexity and running time. WL-GNN [1] and KerGNN [2] are employed as the representative GNNs on modeling high-order relationships. Their computational complexities are between $O(n^2)$ and $O(n^3)$ for capturing and modeling high-order relationship, and the running time of them are shown in Table R1.
> >
> > Table R1. Running time in seconds.
> > |Method|Cora|Pubmed|Citeseer| Comput| Photo| CS| Physics|
> > |------|---:|-----:|-----:|-----:|-----:|-----:|-----:|
> > |GCN|9.89|6.23|5.32|16.8|6.59|19.2|21.58|
> > |GAT|10.45|49.31|12.85|95.23|42.11|106.06|201.79|
> > |OPEN|12.72|45.02|15.85|83.82|44.84|107.2|202.31|
> > |WL-GNN|27.72|197.66|35.95|962.95|852.91|OOM|OOM|
> > |KerGNN|30.88|OOM|95.81|2536.03|1391.74|OOM|OOM|
> >
> >
> > Thus, OPEN is more efficient than other GNNs, which high-order relationships. Besides, we demonstrated that OPEN is as efficient as GNNs modeling pairwise relationships, e.g. GAT, in the previous response (see R1). Therefore, OPEN is high efficient as high-order GNN.
> >
> > >  Q3+. The issues solved by OPEN.
> >
> > R3+. OPEN solves both over-fitting and over-smoothing issues, while Ortho-GConv only solves the over-smoothing issue.
> >
> >
> > **Overfitting Issue**
> >
> > - **How does OPEN solve the overfitting issue?**
> >   The overfitting issue is caused by the over-complicated models, which contain a large number of parameters to be learned [3]. Constrained GAT (C-GAT) demonstrates that GAT has an over-fitting issue, since the attention mechanism introduces additional parameters in learning propagation weights from labels [4]. The intuitive way to solve the over-fitting is to reduce the number of parameters during learning propagation weights. To this end, OPEN proposes to infer the propagation weight from attributes, without learning additional parameters (Line 173-175). Therefore, the whole algorithm of OPEN achieves preventing the overfitting issue.
> >
> > - **How to verify that the overfitting issue is solved?**
> >   Following the Constrained GAT (C-GAT) [4], the ability of OPEN on preventing the overfitting issue is measured via its robustness to noise (Section 4.7 and Figure 4). Compared to the irrelevant propagation model GAT, the proposed relevant propagation model OPEN is more robust to noises as shown in Figure 4. Therefore, the ability of OPEN on preventing the overfitting issue is experimentally verified.
> >
> >  **Over-smoothing Issue**
> >
> > - **How does OPEN solve the over-smoothing issue?**
> >   OPEN is NOT originally designed to solve the over-smoothing issue. Actually, both experimental observations and theoretical analysis reveal that its characteristic of preventing overfitting can be ascribed to the propagation weight inference from ego-network attribute. Considering that, the over-smoothing issue, which is described as the information loss of original node attribute during the embedding, is caused by repeatedly performing topology on the attribute. In this aspect, the proposed OPEN refines the topology with original attributes via ego-network modeling, therefore tackles the over-smoothing issue.
> >
> >
> > - **How to verify that the over-smoothing issue is solved?**
> >   It is verified from both theoretical and experimental perspectives. Theoretical analysis (Theorem 1 in Section 3.4) demonstrates that final embeddings from OPEN contain the attribute information of the ego-network. Experimental evaluations (Section 4.6 and Figure 3) demonstrate that the performances of OPEN do not remarkably drop after many layers. Therefore, its ability on preventing the over-smoothing issue is verified.
> >
> > In conclusion, OPEN solves the overfitting issue from the original design and the over-smoothing issue from experiential observations and the theoretical analysis. Therefore, the proposed OPEN solves more issues than Ortho-GConv and is capable of learning deep and lightweight models.
> >
> >
> > [1] Christopher Morris, Martin Ritzert, Matthias Fey, William L. Hamilton, Jan Eric Lenssen, Gaurav Rattan, Martin Grohe: Weisfeiler and Leman Go Neural: Higher-Order Graph Neural Networks. AAAI 2019: 4602-4609
> >
> > [2] Aosong Feng, Chenyu You, Shiqiang Wang, Leandros Tassiulas: KerGNNs: Interpretable Graph Neural Networks with Graph Kernels. AAAI 2022: 6614-6622
> >
> >
> > [3] Trevor Hastie, Robert Tibshirani, Jerome H. Friedman: The Elements of Statistical Learning: Data Mining, Inference, and Prediction, 2nd Edition. Springer Series in Statistics, Springer 2009, ISBN 9780387848570, pp. I-XXII, 1-745
> >
> > [4] Guangtao Wang, Rex Ying, Jing Huang, Jure Leskovec: Improving Graph Attention Networks with Large Margin-based Constraints. CoRR abs/1910.11945 (2019)

---

### Official Review · Reviewer_GFKV · 2022-07-15

**Rating:** 5
**Confidence:** 1
**Soundness:** 2 fair
**Presentation:** 1 poor
**Contribution:** 2 fair

**Summary:**

The paper tackles the problem of irrelevant propagation to one node and in multi-channels in GNNs via modeling relevances between propagations. The proposed method is called OPEN for Orthogonal Propagation with Ego-Network modeling. The basic idea relies on dimension reduction through principal component analysis to infer the propagation weights. Theoretical and empirical evidence suggests the effectiveness of the proposed method.

**Questions:**

The proposed method relies mainly on principal component analysis, while many other dimension reduction techniques exist in the literature (e.g., kernel PCA). The authors should elaborate more on this aspect by comparing different dimension reduction strategies and comparing them to the proposed scheme.

**Ethics Review Area:**

["I don’t know"]

**Limitations:**

The theoretical analysis (section 3.4) is not consistent. Indeed, the intuitions behind Theorem 1 are not clear and should be clarified more precisely by the authors. It would be more interesting to provide a theoretical quantitative evaluation of the proposed method to prove its effectiveness in theory.

**Strengths And Weaknesses:**

- The paper studies one of the main issues in GNNs; irrelevant propagation and proposes a simple yet effective method to tackle this issue.
- Extensive empirical evaluations are carried out and show the effectiveness of the proposed method.

---

> ### Author Response · Authors · 2022-08-02
> **Response to Reviewer GFKV**
>
> > Q1. The proposed method relies mainly on principal component analysis, while many other dimension reduction techniques exist in the literature (e.g., kernel PCA). The authors should elaborate more on this aspect by comparing different dimension reduction strategies and comparing them to the proposed scheme.
>
> R1. Most unsupervised dimensionality reduction techniques, including PCA, KPCA, MDS, ISOMAP and EIgenmap etc., can be employed as the ego-network modeling, since the proposed OPEN learns node representations via dimensionality reduction on ego-network. The differences between dimensionality reduction techniques lie in their assumptions on data distribution and the construction of similarity matrix. PCA is employed for three reasons. 1) its simplicity and universality; 2) its assumption that data follows Gaussian distribution; 3) orthogonality of its mapping directions, which facilitates the diversity requirements for different propagation channels. Kernel PCA (KPCA) enhances PCA by utilizing the kernel trick in computing covariance matrix $S$, and thus can be employed by the proposed OPEN via replacing the similarity matrix. Similarly, MDS, ISOMAP and EIgenmap also seek the mapping direction via the Eigen-decomposition of (different) similarity matrix. Their performances are provided as follows.  They achieve the similar performance as OPEN.
>
> Table R1. Performance comparison of OPEN with different dimension reduction method.
> |Method       |   Cora  |  Pubmed | Citeseer  |   Comput | Photo  | CS | Physics|
> |-----------|:--------:|:---------:|:-----------:|:----------:|:--------:|:----:|:--------:|
> |OPEN-PCA    |    89.31  | 90.05  | 80.42 |  90.93  | 95.43  | 94.99 | 96.92|
> |OPEN-KPCA    |   89.52  | 90.11  | 80.35 | 90.88  | 95.59  | 95.04 | 96.98|
> |OPEN-MDS      |  89.13  | 89.75  | 80.22 | 90.79  | 95.44  | 94.71 | 96.51 |
> |OPEN-ISOMAP   |  89.31  | 90.05  | 80.42 | 90.93  | 95.43  | 94.99 | 96.92|
>
>
>
>
> > Q2. The theoretical analysis (section 3.4) is not consistent. Indeed, the intuitions behind Theorem 1 are not clear and should be clarified more precisely by the authors. It would be more interesting to provide a theoretical quantitative evaluation of the proposed method to prove its effectiveness in theory.
>
> R2. The intuition behind Theorem 1 is that over-smoothing issue can be alleviated by refining the propagation weights via the ego-network’s attribute. The over-smoothing phenomenon is that nodes lose their original attribute information in the representations, thus become indistinguishable from each other. As claimed by pervious works, the over-smoothing issue is caused by repeatedly propagating node attribute via topology-wise weights, e.g., node degree in Eq. (11). The proposed OPEN represents node via its corresponding ego-network modeling, which is equivalent to refining the propagation weight via the ego-network’s attribute, thus alleviates the degradation and the over-smoothing. Theorem 1 demonstrates that node representation obtained from OPEN is relevant to the principal components of its corresponding ego-network’s attribute, thus they can be distinguished from each other. Therefore, OPEN can alleviate over-smoothing issue by refining the propagation weights via the ego-network’s attribute.
>
> During the rebuttal, as you suggested, we try to find a theoretical quantitative evaluation to prove its effectiveness in theory. However, we find it is very difficult, since the representation obtained from OPEN is a complicated combination of topology and node attribute. To make it more rigorous, Theorem 1 is refined as follows.
>
> Theorem 1. The representation of each node from OPEN is relevant to the principal components of its corresponding ego-network’s attribute.

---

> ### Author Response · Authors · 2022-08-07
> **If you have any more questions, we are happy to discuss them**
>
> Dear Reviewer GFKV, we sincerely appreciate the time and efforts you have given to our paper's review. We hope that you can kindly consider our responses in making the final decision on our paper. The reviewer-author discussion period is going to end soon. Please let us know whether our responses have adequately addressed your questions and concerns. If you have any more questions, we are happy to discuss them.

---

> > ### Comment · Reviewer_GFKV · 2022-08-08
> > **Score changed to 5**
> >
> > Thank you for addressing my comments. I changed my score to borderline accept since I believe only Q1 has been addressed and Q2 about the theoretical guarantees of the proposed method is still not convincing for me.

---

> > > ### Author Response · Authors · 2022-08-09
> > > **Sincere appreciation for your insightful comments and compliments**
> > >
> > > We would like to express our sincere appreciation to you for your insightful comments and compliments to our paper.

---

### Meta-Review · Area_Chair_3LA1 · 2022-08-27

**Recommendation:** Accept
**Confidence:** Less certain

**Metareview:**

Reviewers recommended borderline accept, borderline reject, and accept. Reviewers found the article studies one of the main issues of GNNs and proposes a simple but effective solution method supported by extensive experimental evaluation. There were some reservations about the comparison with other methods and the theoretical analysis. While some of these items could be addressed during the discussion period, leading to updated more favorable ratings, some reservations about the theoretical part persisted. There were also persisting disagreements about the novelty and about the issues that are solved by the proposed method compared with previous methods. All together I found that the merits outweighed the shortcomings and hence am recommending accept. However, I strongly encourage the authors to carefully consider the reviewers comments when preparing the final manuscript, particularly that they work on the discussion and clarification of the novelty of the method, particularly the issues between over smoothing and overfitting and the corresponding presentation in the work, and the reservations on the theoretical part.

**Award:**

No

---

### Decision · Program_Chairs · 2022-09-14

Accept